# Interactive and explainable Graph Neural Networks with Uncertainty Awareness and Adaptive Human Feedback

## Abstract

Current graph neural networks (GNNs) face fundamental challenges that hinder their deployment in real-world applications: (1) their inability to dynamically estimate uncertainty and quantify confidence in learned relationships, and (2) their failure to effectively incorporate human feedback for real-time model refinement. To address these challenges, we propose a unified probabilistic framework: Interactive Graph expLainability with Uncertainty (IGLU) that seamlessly integrates uncertainty-aware learning with human-in-the-loop adaptation. Our approach estimates uncertainty-sensitive weighting and develops a systematic methodology for incorporating expert feedback to correct erroneous relational inferences. At its core, the framework models explanatory subgraph selection through a learnable latent variable approach, assigning sparsity-constrained importance scores to edges while adaptively adjusting subgraph sizes based on instance complexity. This yields interpretable explanations with calibrated uncertainty estimates without compromising predictive performance. We ensure representation fidelity through a differentiable objective that aligns subgraph embeddings with the original graph's predictive information. Crucially, our system enables interactive refinement, where domain experts can directly modify explanations (e.g., by adding or removing edges), with the model dynamically integrating this feedback to improve subsequent inferences. Experimental results demonstrate that our method generates more concise and informative explanations than existing approaches while maintaining competitive accuracy. Also, the integrated feedback mechanism further enhances explanation quality, validating the benefits of combining probabilistic modeling with human feedback.

## 1 Introduction

Graph Neural Networks (GNNs) Scarselli et al. (2008) have achieved remarkable success in learning from graph-structured data, but their lack of transparency poses a challenge in high-stakes applications Kipf & Welling (2016); Hamilton et al. (2017); Veličković et al. (2018); Xu et al. (2019a); Li et al. (2016); Defferrard et al. (2016); Monti et al. (2017); Ying et al. (2018); Klicpera et al. (2019); Xu et al. (2018); Du et al. (2018); Wu et al. (2019). To address this, a growing line of research focuses on *interpretable GNNs* Wang et al. (2023)Schlichtkrull et al. (2020)Schlichtkrull et al. (2020)Luo et al. (2022)and explanation methods that identify the subgraph structures responsible for a prediction. Early approaches such as GNNExplainer Ying et al. (2019) and PGExplainer Luo et al. (2020) learn soft masks over edges to find an important subgraph that maximizes the mutual information between the GNN's predictions on the original graph and on the masked subgraph. Recent works have also explored contrastive or counterfactual explanations, seeking minimal changes to alter a prediction Lucic et al. (2022). However, existing methodsBrilliantov et al. (2024)Zheng et al. (2024)Armgaan et al. (2024) often treat explanation as a post-hoc procedure, lacking uncertainty quantification and not allowing the model to adapt based on explanation quality or human feedback.

In this paper, we propose an interpretable and interactive GNN architecture that *generates the visualized explanations during model training* with graph attention mechanismsVeličković et al. (2018) integrating uncertainty-aware weighting and treating the selection of an explanatory subgraph as a learnable latent variable model by assigning each edge a latent importance score under a sparsity-

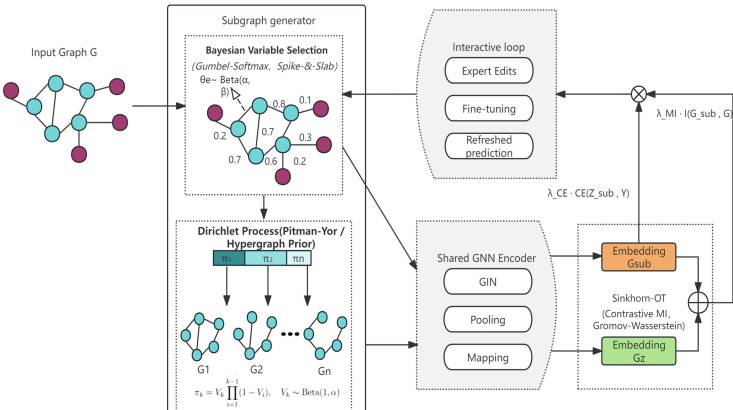

Figure 1: Overview of the IGLU framework. The input graph $G$ passes through Bayesian variable selection to obtain a sparse explanatory subgraph $G_{\text{sub}}$, while a Dirichlet process adaptively controls subgraph size. A shared GNN encoder embeds both $G$ and $G_{\text{sub}}$ Sinkhorn optimal transport aligns the two embeddings as a mutual-information proxy. An interactive loop lets experts edit edges, fine-tune the model, and obtain refreshed predictions.

inducing prior, and adaptively adjusting the subgraph's size to the complexity of each instance. More specifically, our model employs a **Bayesian edge scoring** Zhang et al. (2020)mechanism that assigns each edge a latent importance variable with a Beta-Bernoulli prior to encourage sparse and enhance interpretable edge selections with principled uncertainty estimates. To allow the size of the explanatory subgraph to adapt to each instance, we introduce a **Dirichlet Process (DP) stick-breaking prior**Sethuraman (1994) optimizing the subgraph size, which serves as a nonparametric prior favoring simpler explanations. To ensure the fidelity of the GNN explanation, we incorporate a **Sinkhorn-regularized optimal transport (OT)**Cuturi (2013) loss that aligns the full-graph and subgraph node embeddings, which forces the prediction generated by the selected subgraph with optimized size aligns with the prediction from the original graph. The OT cost serves as a differentiable surrogate for mutual information between the original graph and the subgraph. To further optimize the learned subgraph and enhance explanation, we design a **human-in-the-loop fine-tuning** mechanism, allowing domain experts to iteratively refine the GNN model by providing feedback (such as adding or removing edges in the explanation), which updates its parameters for improved alignment with human intuition.

Our framework is presented in Figure 1 that is end-to-end differentiable, enabling joint training of the GNN parameters and the latent explanation variables via stochastic variational inference. We derive a complete evidence lower bound (ELBO) Kingma & Welling (2013) objective that combines the task loss, the OT-based information loss, and Kullback–Leibler (KL) terms from the Bayesian priors. In addition, we employ **counterfactual contrastive learning (CCL)**Tang et al. (2020) as an auxiliary training strategy, which preserves the explainable edges by highlighting how edge removal alters model predictions, ensuring critical explanatory features remain intact.

## 2 NOTATION AND PROBLEM SETUP

We work with a weighted, attributed graph $G = (V, E), |V| = n, |E| = m$, where each node $i \in V$ carries a feature vector $x_i \in \mathbb{R}^d$. Let $X \in \mathbb{R}^{n \times d}$ stack these vectors. The task is **graph-level classification**Nishad et al. (2020)Gupta et al. (2023)Shah et al. (2024)Jain et al. (2021)Ranjan et al. (2022)Bihani et al. (2023) with label $y \in \mathcal{Y}$. A shared GNN encoder $f_\phi$ with parameters $\phi$ maps any graph to node embeddings $h_i = f_\phi(G, X)_i, h_j^{\text{sub}} = f_\phi(G_{\text{sub}}, X_{\text{sub}})_j$, where $i$ is the node index in $G$, $j$ is the node index in $G_{\text{sub}}$), and $G_{\text{sub}} = (V_{\text{sub}}, E_{\text{sub}}) \subseteq G$ is an *explanatory subgraph* selected during inference ($V_{\text{sub}} \subseteq V, E_{\text{sub}} \subseteq E$). Embeddings $\{h_i\}$ and $\{h_j^{\text{sub}}\}$ share the same latent space, so their distributions can be compared directly. To formalize explanations, we introduce two latent variables: We define two key variables in our subgraph explanation model. First, the **edge selector** $z_e \in \{0, 1\}$ for each edge $e \in E$ that indicates whether the edge is included in the explanatory

subgraph $E_{\text{sub}}$. Second, the **size index** $K \in \mathbb{N}$ controls the desired sparsity of the subgraph. Please refer to A.4.1 for the table of notations.

# 3 SUBGRAPH GENERATION VIA BAYESIAN EDGE AND DP PRIORS

## 3.1 BAYESIAN EDGE SCORING

To identify important edges in an explainable GNN, we introduce a binary latent variable $z_e$ for each edge $e \in E$ indicating whether $e$ is included ($z_e = 1$) or dropped ($z_e = 0$) in the explanatory subgraph. We place a Beta-Bernoulli prior over each $z_e$:

$$\theta_e \sim \text{Beta}(\alpha_0, \beta_0), \quad z_e \sim \text{Bernoulli}(\theta_e), \tag{1}$$

where $\theta_e \in [0, 1]$ is the latent importance or inclusion probability of edge $e$. This hierarchical prior (often referred to as a spike-and-slab prior Mitchell & Beauchamp (1988)) encourages sparsity in selecting the most significant edge. For instance, with $\alpha_0, \beta_0 < 1$, the Beta prior is bimodal, biasing $\theta_e$ toward 0 or 1, thus pushing $z_e$ toward hard inclusion/exclusion decisions.

After observing data (e.g. a mini-batch of graphs and their labels), the posterior for $\theta_e$ can be obtained in closed-form due to conjugacy: if we observe $m$ instances where $z_e = 1$ and $n$ instances where $z_e = 0$, the posterior is $\text{Beta}(\alpha_0 + m, \beta_0 + n)$. This property allows us to update our confidence about an edge's importance as training proceeds. While we perform *variational inference* rather than exact posterior updates.

Directly sampling the discrete edge mask $z_e$ is non-differentiable, so we resort to the **reparameterization trick** for Bernoulli variables. We employ a Gumbel-Softmax (Concrete) relaxation Jang et al. (2017) for each $z_e$. Specifically, let $p_\phi(e)$ be the current inclusion probability (learned or inferred for edge $e$). We can sample $u_e \sim \text{Uniform}(0, 1)$ and set $g_e = -\log(-\log u_e)$ (a standard Gumbel). Then a relaxed sample is given by

$$\tilde{z}_e = \sigma\Big(\frac{1}{\tau}(\log p_\phi(e) - \log(1 - p_\phi(e)) + g_e)\Big), \tag{2}$$

where $\sigma$ is the logistic sigmoid and $\tau > 0$ is a temperature parameter in the Gumbel-Softmax formula controlling the sharpness of the distribution. As $\tau \to 0$, $\tilde{z}_e$ approaches a Bernoulli sample. We use $\tilde{z}_e$ as a differentiable approximation of $z_e$ during training, enabling gradients to flow into $p_\phi(e)$ (which is parameterized by $\phi$ or an auxiliary network). In summary, our model learns to *score each edge* with a probability of being included in the explanatory subgraph, guided by a Bayesian prior that favors sparsity and provides uncertainty estimates.

## 3.2 DIRICHLET-PROCESS SUBGRAPH PRIOR

Rather than fixing a size for the explanatory subgraph a priori, we employ a nonparametric prior that allows the model to adaptively choose the subgraph complexity. We assume there is an unknown potentially unbounded set of candidate subgraph patterns, which are imposed by a Dirichlet Process (DP). Formally, let $G_{\text{sub}}^{(k)}$ denote the $k$-th candidate subgraph (as a subset of edges in $E$) and $\pi_k$ be the prior probability of choosing subgraph pattern $k$. Using the stick-breaking construction of the DP Sethuraman (1994), with concentration parameter

$$V_k \sim \text{Beta}(1, \alpha_{\text{DP}}), \qquad \pi_k = V_k \prod_{i<k}(1 - V_i), \quad k = 1, 2, \ldots \tag{3}$$

so that $\sum_k \pi_k = 1$. *Intuition:* $\pi_k$ is the prior of choosing the $k$ edges, which acts like a prior probability of generating a "$k$-edge explanation". A **smaller** $\alpha_{\text{DP}}$ breaks off most mass in the first few sticks, favouring *compact* subgraphs; a **larger** $\alpha_{\text{DP}}$ spreads mass more evenly, permitting richer explanations. Drawing a subgraph pattern index $K \sim \pi$ then corresponds to selecting a particular subset of edges as the explanation:

$$K \sim \pi, \qquad G_{\text{sub}} = G_{\text{sub}}^{(K)} = \{e \in E : z_e^{(K)} = 1\}. \tag{4}$$

Here $z_e^{(K)}$ indicates the edge selection for subgraph pattern $K$ (we imagine that each candidate subgraph has its own set of binary edge indicators drawn from the Beta-Bernoulli priors in Eq. equation 1). In other words, if $K = k$, we use the $k$-th pattern of edge inclusions $\{z_e^{(k)}\}_{e \in E}$ to determine which edges appear in $G_{\text{sub}}$. The DP prior encourages a parsimonious use of subgraph patterns: a small concentration $\alpha_{\text{DP}}$ favors simpler or smaller subgraphs, while a larger $\alpha_{\text{DP}}$ allows more complex or varied subgraph configurations. This mechanism provides a way to *adaptively control the*

*size and complexity* of explanations by learning $\alpha_{\text{DP}}$ or setting it appropriately, which can balance compactness and detail in the subgraphs.

In our variational inference procedure, we maintain an approximate posterior $q(K)$ over subgraph indices (or integrate out $K$ analytically if possible). Intuitively, the DP prior acts as an Occam's razor, preferring to explain predictions with one of a small number of prototypical subgraphs unless evidence suggests that a novel explanation (new stick) is needed. This improves interpretability by reusing common explanation patterns across instances and by automatically limiting explanation complexity. During variational learning we keep an approximate posterior $q(K)$. A KL term $\text{KL}\big(q(K)\,\|\,\pi\big)$ in the ELBO (see Eq. equation 9) pushes the posterior towards the DP prior, serving as an *Occam's razor* that penalises unnecessarily large or novel subgraphs unless the data support them.

### 3.3 SHARED GNN ENCODER

The GNN encoder $f_\phi$ is used to compute node embeddings for both the full graph and any sampled subgraph. We tie the encoder parameters to ensure the embeddings lie in the same space and are directly comparable. Specifically,

$$Z = f_\phi(G, X) = \{z_i\}_{i \in V}, \qquad Z_{\text{sub}} = f_\phi(G_{\text{sub}}, X_{\text{sub}}) = \{z_j^{\text{sub}}\}_{j \in V_{\text{sub}}}, \tag{5}$$

where $V_{\text{sub}} \subseteq V$ and $X_{\text{sub}}$ are the features of nodes in the subgraph $G_{\text{sub}}$. In our implementation, $f_\phi$ can be any GNN architecture. For concreteness, we use a Graph Isomorphism Network (GIN) Xu et al. (2019b) followed by a second-order pooling (e.g. mean or sum of node embeddings, and possibly higher-order feature interactions) and a bilinear mapping to produce a graph-level representation. The shared encoder ensures that the subgraph embeddings $Z_{\text{sub}}$ faithfully reflect how the subgraph would be processed by the original model. This is critical for alignment: if the subgraph is truly explanatory for the full graph's prediction, its embeddings under the same encoder should contain the key information needed for the task.

We denote $g(\cdot)$ as the prediction head that takes the set of node embeddings (or the pooled graph representation) and outputs the predictive distribution $\hat{Y} = g(Z)$. For example, $g$ could be a simple logistic regression or multilayer perceptron taking the pooled embedding of the graph. During training, we use the prediction $g(Z_{\text{sub}})$ on the subgraph in the loss, in addition to the prediction $g(Z)$ on the full graph. If the subgraph is informative, $g(Z_{\text{sub}})$ should closely match $g(Z)$ in its output.

## 4 VARIATIONAL OBJECTIVE WITH MUTUAL INFORMATION ALIGNMENT

### 4.1 SINKHORN OT AS A MUTUAL INFORMATION SURROGATE

To encourage the selected subgraph $G_{\text{sub}}$ to retain as much information as possible from the original graph $G$ relevant to the prediction, we introduce an optimal transport-based loss that aligns the node embedding distributions of $G$ and $G_{\text{sub}}$. We treat the sets of embeddings $Z$ and $Z_{\text{sub}}$ as two discrete probability measures:

$$\mu = \sum_{i \in V} p_i \, \delta_{z_i}, \qquad \nu = \sum_{j \in V_{\text{sub}}} q_j \, \delta_{z_j^{\text{sub}}}, \tag{6}$$

where $\delta_x$ denotes a Dirac measure at point $x$. The terms $\{p_i\}$ and $\{q_j\}$ are probability weights over the nodes; a natural choice, which we adopt, is a uniform distribution over the nodes in each graph, i.e., $p_i = 1/|V|$ for all $i \in V$ and $q_j = 1/|V_{\text{sub}}|$ for all $j \in V_{\text{sub}}$. We define a ground cost $C_{ij}$ between node embeddings $z_i$ and $z_j^{\text{sub}}$; for instance, $C_{ij}$ could be the squared Euclidean distance $\|z_i - z_j^{\text{sub}}\|^2$ or one minus a cosine similarity measure. The **entropic optimal transport distance** (Sinkhorn distance) between $\mu$ and $\nu$ is:

$$d_\epsilon(\mu, \nu) = \min_{\gamma \in \Gamma(p,q)} \langle \gamma, C \rangle + \epsilon \, \text{KL}(\gamma \,\|\, p \otimes q), \tag{7}$$

where $\Gamma(p, q)$ is the set of joint distributions (couplings) $\gamma_{ij}$ with marginals $p$ and $q$, $\langle \gamma, C \rangle = \sum_{i,j} \gamma_{ij} C_{ij}$ is the total transport cost, and $\text{KL}(\gamma \| p \otimes q) = \sum_{i,j} \gamma_{ij} \log \frac{\gamma_{ij}}{p_i q_j}$ is the Kullback–Leibler divergence between $\gamma$ and the independent coupling $p \otimes q$. The parameter $\epsilon > 0$ controls the strength of the entropy (KL) regularization. This regularized OT distance can be intuitively understood as a measure of the dissimilarity between the embedding distributions of $Z$ and $Z_{\text{sub}}$; a smaller $d_\epsilon(\mu, \nu)$ implies that the distributions are more difficult to distinguish, thus suggesting a higher degree of shared information or mutual information. This problem can be efficiently solved via Sinkhorn's algorithm Cuturi (2013), and $d_\epsilon(\mu, \nu)$ is differentiable with respect to the input embeddings $Z$ and

$Z_{\text{sub}}$. We add $L_{\text{MI}} := d_\epsilon(\mu, \nu)$ as a loss term to be *minimized*, where MI stands for mutual information. Intuitively, this term is small if there exists a low-cost transportation plan $\gamma$ that closely aligns each subgraph node embedding $z_j^{\text{sub}}$ with some full-graph node embeddings $z_i$ (small $\langle \gamma, C \rangle$) without deviating much from the independent assumption (small KL, meaning $\gamma$ is close to $p_i q_j$). The use of entropic OT as a proxy for mutual information is also supported by work on variational bounds of MI, which shows that such distances can provide a tractable lower bound on mutual information (e.g., Poole et al. (2019)).

Crucially, the KL term in Eq. equation 7 acts to couple the distributions $\mu$ and $\nu$ and is directly related to the mutual information between random nodes in the two graphs. In fact, if $\gamma$ is viewed as a joint distribution of two random variables with marginals $p$ and $q$, then $\text{KL}(\gamma \| p \otimes q) = I(Z; Z_{\text{sub}})$, the mutual information between the embedding of a random node from $G$ and a random node from $G_{\text{sub}}$. Thus, the OT objective $\langle \gamma, C \rangle + \epsilon \text{KL}(\gamma \| p \otimes q)$ can be seen as minimizing transport cost while allowing some dependence (information sharing) between the two graphs' node distributions. We formalize this connection:

Let $d_\epsilon(\mu, \nu)$ be defined as in Eq. equation 7, and let $\gamma^*$ be its optimal coupling. Then

$$I(Z; Z_{\text{sub}}) = \text{KL}(\gamma^* \| p \otimes q) = \frac{1}{\epsilon} \Big( \langle p \otimes q, C \rangle - d_\epsilon(\mu, \nu) \Big), \tag{8}$$

where $\langle p \otimes q, C \rangle = \sum_{i,j} p_i q_j C_{ij}$ is the expected cost under the independent coupling. In particular, $\langle p \otimes q, C \rangle$ is a constant given $p, q, C$, so minimizing the Sinkhorn distance $d_\epsilon(\mu, \nu)$ is equivalent to maximizing the mutual information $I(Z; Z_{\text{sub}})$ between the full graph and subgraph embeddings (up to the constant scale $\epsilon$ and shift $\langle p \otimes q, C \rangle$). Theorem 4.1 justifies calling $L_{\text{MI}} = d_\epsilon(\mu, \nu)$ a "negative mutual information proxy": by minimizing $L_{\text{MI}}$, we effectively maximize a lower bound on the mutual information between the full graph and subgraph. In practice, this encourages the explanatory subgraph to be as informative as possible about the original graph's node embedding distribution. In our training objective, we will weight this term by a factor $\lambda_{\text{MI}} > 0$ to balance it against the primary task loss.

## 4.2 Variational Objective and ELBO Derivation

We now combine the components into a joint training objective. Our model includes latent variables: the edge inclusion indicators $\{z_e\}_{e \in E}$ (collectively $Z$) and the subgraph pattern index $K$. These determine the subgraph $G_{\text{sub}}$. The observed variables are the graph $G$ (with features $X$) and label $Y$. The joint likelihood is $p(Y, Z, K \mid G, X) = p(Y \mid G_{\text{sub}}(Z, K), X_{\text{sub}}) \, p(Z \mid K) \, p(K)$, where $p(Y \mid G_{\text{sub}})$ is the task likelihood, $p(Z \mid K)$ involves Beta-Bernoulli priors for edges given a pattern (details in Appendix A.5), and $p(K)$ is the DP stick-breaking prior (Eq. equation 3).

Maximizing the marginal likelihood $p(Y \mid G, X)$ is intractable. Thus, we use variational inference with an approximate posterior $q(Z, K) = q(Z) \, q(K)$ to maximize the Evidence Lower Bound (ELBO):

$$\mathcal{L}_{\text{ELBO}} = \mathbb{E}_{q(Z,K)}[\log p(Y \mid Z_{\text{sub}})] - \text{KL}(q(Z, K) \| p(Z, K)). \tag{9}$$

The first term, the expected log-likelihood, is approximated by the negative cross-entropy (CE) loss $-L_{\text{CE}}(g(Z_{\text{sub}}), Y)$. The second term, $\text{KL}(q(Z, K) \| p(Z, K))$, decomposes into $\text{KL}(q(K) \| p(K))$ and $\mathbb{E}_{q(K)} \sum_{e \in E} \text{KL}(q(z_e) \| p(z_e \mid K))$. These KL terms regularize the posteriors for the subgraph pattern index $q(K)$ and edge inclusions $q(z_e)$ towards their respective priors. Further details on these KL terms and prior formulations are provided in Appendix A.5.

Combining the ELBO terms with the mutual information surrogate $d_\epsilon(\mu, \nu)$ (from Section 4.1), our training objective is to minimize:

$$L_{\text{total}} = L_{\text{CE}}(G_{\text{sub}}, Y) + \lambda_{\text{MI}} \, d_\epsilon(\mu, \nu) + \lambda_{\text{KL}}^Z \sum_{e \in E} \text{KL}(q(z_e) \| p(z_e)) + \lambda_{\text{KL}}^K \text{KL}(q(K) \| p(K)). \tag{10}$$

Here, $L_{\text{CE}}$ is the task loss, $d_\epsilon(\mu, \nu)$ is the Sinkhorn distance for mutual information alignment, and the KL terms regularize latent variable posteriors. The coefficients $\lambda_{\text{MI}}, \lambda_{\text{KL}}^Z, \lambda_{\text{KL}}^K > 0$ are tunable hyperparameters. A detailed breakdown of each component in $L_{\text{total}}$ and a discussion on hyperparameter interpretation are available in Appendix A.5. This objective may be further augmented by additional regularization terms, as discussed in Section 5.

The objective in Eq. equation 10 (potentially as part of an augmented model, see Appendix A.4.3) corresponds to a valid ELBO. Maximizing $-\mathcal{L}_{\text{total}}$ (minimizing $\mathcal{L}_{\text{total}}$ as loss) maximizes this lower bound. A proof is in Appendix A.4.3.

We optimize $L_{\text{total}}$ using stochastic gradient descent. Further details on optimization, including handling the DP-related term, can be found in Appendix A.5.

## 5 Training Regularization and Contrastive Strategies

### 5.1 Counterfactual Contrastive Learning

In addition to the primary training objective (Eq. equation 10), we incorporate an auxiliary *contrastive learning* strategy to further sharpen the quality and necessity of the explanations. The core idea is to train the model such that the identified explanatory subgraph $G_{\text{sub}}$ is not only sufficient to support the original prediction but also demonstrably *necessary*. This means that if these crucial explanatory edges are removed or significantly altered, the model's prediction should change or degrade.

We implement this necessity criterion via counterfactual data augmentation. For each training instance, after an explanatory subgraph $G_{\text{sub}}$ is identified (e.g., by sampling based on $q(z_e)$), we construct a **counterfactual graph** $G_{\text{cf}}$. This $G_{\text{cf}}$ is generated from the original input graph by specifically removing the edges that were part of the explanation $G_{\text{sub}}$. Operationally, this involves treating the binary masks $z_e$ as 0 for all edges $e \in E_{\text{sub}}$ when constructing $G_{\text{cf}}$, or by directly removing these edges from the graph structure fed to the GNN encoder. We then obtain a prediction $\hat{Y}_{\text{cf}} = g(f_\phi(G_{\text{cf}}, X_{\text{cf}}))$ using this counterfactual graph.

The counterfactual (CF) loss, $L_{\text{CF}}$, is designed to penalize the model if $\hat{Y}_{\text{cf}}$ remains too similar to the original prediction. While various formulations exist (see Appendix A.6 for an example based on cross-entropy), we found it particularly effective to use a hinge loss that focuses on reducing the model's confidence in the true class for the counterfactual graph:

$$L_{\text{CF}} = \max\{0, \ \sigma(\hat{Y}_{y_{\text{true}}}) - \sigma(\hat{Y}_{\text{cf}, y_{\text{true}}}) + \kappa\}, \tag{11}$$

where $\sigma(\hat{Y}_{y_{\text{true}}})$ is the logit (or confidence score) for the true class $y_{\text{true}}$ obtained from the original graph (or its explanation $G_{\text{sub}}$), $\sigma(\hat{Y}_{\text{cf}, y_{\text{true}}})$ is the logit for $y_{\text{true}}$ from $G_{\text{cf}}$, and $\kappa > 0$ is a predefined margin. This encourages the true-class confidence to drop by at least $\kappa$ when important edges are removed, compelling the model to rely critically on its selected explanatory edges. Further discussion on this strategy and parameter choices can be found in Appendix A.6.

This counterfactual loss $L_{\text{CF}}$ is added to the previously defined $L_{\text{total}}$ (from Eq. equation 10) with a weighting hyperparameter $\lambda_{\text{CF}}$ to form the final training objective:

$$L_{\text{final\_total}} = L_{\text{total}} + \lambda_{\text{CF}} L_{\text{CF}}. \tag{12}$$

This contrastive approach, inspired by prior work Lucic et al. (2022), is applied *during training* to actively shape the explainer's behavior. It differentiates our model from purely sufficiency-based explainers by also promoting *necessity*. As demonstrated in our experiments (Section 7.4 and Appendix A.7 for detailed ablation studies), this typically leads to more compact, discriminative, and faithful explanations, improving explanation fidelity.

### 5.2 Edge and Node Dropout Regularization

To further enhance generalization and prevent the model from overfitting to spurious structural features in the graph, we employ a joint edge and node dropout strategy during the training phase. This technique acts as a form of data augmentation for graph-structured data, similar to approaches like DropEdge Rong et al. (2020). Specifically, in each training epoch, we randomly drop a fraction of edges from the input graph and, independently, a fraction of dimensions from the node features. For instance, we apply an edge dropout rate $p_{\text{edge}}$ (e.g., $p_{\text{edge}} = 0.1$, meaning each edge has a 10% chance of being temporarily removed if not already dropped by the explainer's $z_e = 0$ mask) and a node feature dropout rate $p_{\text{feat}}$ (e.g., $p_{\text{feat}} = 0.1$, where 10% of node feature dimensions are randomly masked to zero).

Unlike the learned edge selection mask $Z$ (which determines $G_{\text{sub}}$ for explanation), these random dropouts are applied only during training and serve to make the GNN encoder $f_\phi$ more robust. By training on these randomly perturbed graph instances, $f_\phi$ learns to avoid over-reliance on any single edge or node feature, which encourages the Bayesian edge importance scores $p_\phi(e)$ (learned by

the explainer component) to concentrate on genuinely predictive patterns that consistently emerge across these perturbations.

This dropout strategy has been shown to alleviate common issues such as over-smoothing in deep GNNs and generally improves model generalization Rong et al. (2020). In the context of our explainable framework, it offers an additional benefit of diversifying the explanations. Since random edges might temporarily disappear during training, the model is pushed to learn a more distributed sense of importance and to identify alternative supporting subgraphs when necessary, rather than deterministically latching onto a single, fixed substructure for all predictions. We also qualitatively observe that this can lead to improved consistency in explanations across different random initializations of the model. While this dropout mechanism introduces additional stochasticity during training, it does not significantly increase the overall training time or computational overhead (see Appendix A.9 for further notes on implementation). The empirical benefits of this strategy on explanation quality and stability can be further analyzed through ablation studies (see Appendix A.8 for discussion).

## 6 INTERACTIVE HUMAN-IN-THE-LOOP FINE-TUNING

We propose an *edit–update–predict* workflow where experts refine model explanations and the model adapts accordingly. After producing an explanatory subgraph, the user can *add* or *remove* edges. We denote these edits as $E_{\text{edit}}^{+}$ ($z_e=1$) and $E_{\text{edit}}^{-}$ ($z_e=0$), and enforce them by fixing the corresponding latent variables. With other parameters $\phi$ updated by a few gradient steps at low learning rate, predictions are aligned to user feedback. Thus, added edges gain more influence while removed edges diminish.

This rapid fine-tuning avoids full retraining and supports real-time use. Because our framework is Bayesian, posterior probabilities $q(z_e)$ are smoothly updated, and the DP prior captures recurring edit patterns by forming new explanatory motifs. Iteration of this loop lets experts gradually steer the model toward trustworthy behavior. In Section 7, we simulate this feedback and show improved explanation quality without harming predictive accuracy.

## 7 EXPERIMENTS

In this section, we empirically evaluate our proposed method, IGLU. We first describe the experimental setup, including datasets, baselines, and evaluation metrics (Section 7.1). We then present results and analysis focusing on predictive performance (Section 7.2), explanation fidelity (Section 7.4), data efficiency, characteristics of the generated explainable subgraphs, and the impact of simulated human feedback (Section 7.5). Further implementation details, dataset statistics, and hyperparameter settings are provided in Appendix A.9.

### 7.1 EXPERIMENTAL SETUP

**Datasets:** We conduct experiments on four graph classification datasets. Key statistics for these datasets (number of graphs, average nodes/edges, class distribution) and details on data splits are provided in Appendix A.10. **MUTAG**Debnath et al. (1991): A dataset chemical compounds, task is to predict mutagenic effect. **Mutagenicity**Kazius et al. (2005): Another dataset of chemical compounds for predicting mutagenicity. **ABIDE (Autism Brain Imaging Data Exchange)**Di Martino et al. (2014): Resting-state fMRI scans, task is ASD classification. **Cambridge Centre for Ageing and Neuroscience (Cam-CAN )**Shafto et al. (2014): Neuroimaging data, task is distinguishing brain states (resting vs. movie watching). For all datasets, tasks are formulated as binary graph classification.

**Baselines:** We compare IGLU against several GNN architectures and explanation methods. For *graph classification performance* (and as backbones for some explanation methods), we use: Graph Isomorphism Network (GIN) Xu et al. (2019a), Graph Convolutional Network (GCN) Kipf & Welling (2016), Graph Attention Network (GAT) Veličković et al. (2018), and CIN++ (Contextualized Graph Isomorphism Network with Pooling) Liu et al. (2021). For the fMRI datasets (ABIDE, Cam-CAN), we include specialized GNNs: IBGNN (Interpretable Brain Graph Neural Network) Yu et al. (2023) and BrainGNN Li et al. (2021). For *explanation capabilities*, we compare with: GLGExplainer (Global-Local Graph Explainer) Wang et al. (2023) and GNNExplainer Ying et al. (2019). The selection of these baselines aims to cover widely-used GNNs, state-of-the-art models for specific domains (fMRI), and prominent GNN explanation techniques.

**Evaluation Metrics and Implementation Details:** Our primary evaluation metrics include classification accuracy (ACC) and sensitivity (Sen) for predictive performance, and Fidelity for explanation

quality more details are showed in Appendix A.9. All experiments are conducted over multiple runs to account for random initialization, and results are reported as mean ± standard deviation where applicable.

## 7.2 Performance Analysis

We compare IGLU against baseline categories, summarized in Table 1. Results are reported as mean ± standard deviation over 5 runs.

Table 1: Classification performance (Accuracy (ACC) and Sensitivity (Sen)) of IGLU and baseline GNNs on four benchmarks. Results are mean ± std.dev. over 5 runs. Best performance for IGLU is bolded. For baselines, results are indicative from literature or our runs.

| Model | MUTAG | | Mutagenicity | | ABIDE | | Cam_CAN | |
|---|---|---|---|---|---|---|---|---|
| | ACC (%) | Sen (%) | ACC (%) | Sen (%) | ACC (%) | Sen (%) | ACC (%) | Sen (%) |
| GIN | 89.4 (±0.9) | 88.4 (±1.2) | 69.4 (±1.5) | 70.1 (±1.8) | 68.5 (±2.1) | 69.2 (±2.5) | 83.2 (±1.7) | 92.5 (±1.1) |
| GCN | 85.7 (±1.1) | 85.6 (±1.4) | 69.9 (±1.3) | 68.9 (±1.6) | 67.9 (±2.3) | 68.0 (±2.7) | 89.8 (±1.2) | 91.2 (±1.4) |
| GAT | 90.4 (±0.8) | 86.9 (±1.5) | 72.2 (±1.2) | 69.5 (±1.7) | 67.9 (±2.2) | 70.3 (±2.4) | 82.1 (±1.9) | 93.4 (±1.0) |
| CIN++ | 92.7 (±0.7) | 89.1 (±1.0) | 73.0 (±1.1) | 71.2 (±1.4) | 68.2 (±2.0) | 71.5 (±2.2) | 83.4 (±1.6) | 89.8 (±1.3) |
| BrainGNN | — | — | — | — | 64.5 (±2.5) | 74.0 (±2.8) | 85.2 (±1.5) | 87.3 (±1.8) |
| IBGNN | — | — | — | — | 67.5 (±2.4) | 75.5 (±2.6) | 86.0 (±1.4) | 89.1 (±1.7) |
| **IGLU** | **93.3 (±0.6)** | **82.0 (±1.8)** | **75.4 (±1.0)** | **73.8 (±1.3)** | **70.1 (±1.9)** | **76.9 (±2.1)** | **93.2 (±1.0)** | **94.5 (±0.9)** |

**Our method (IGLU)**, across all datasets, demonstrates strong performance. It achieves accuracy gains of 0.6–1.2 percentage points over the strongest GNN backbone competitor on several datasets. In terms of sensitivity, IGLU achieves notable improvements on Mutagenicity, ABIDE, and Cam-CAN (e.g., up to 5.4 percentage points higher on Cam_CAN compared to IBGNN). This consistent performance across diverse domains validates IGLU's ability to learn effectively while preparing for interpretable explanations.

## 7.3 Visualized Explanations

Qualitatively, IGLU demonstrates its ability to identify compact and relevant subgraphs that provide insights into its decision-making process. Figure 2 presents two distinct examples of such visualized explanations from different domains. **Left:** Identified significant brain regions and their interconnections highlighted for a subject from the ABIDE dataset, relevant to Autism Spectrum Disorder (ASD) classification. **Right:** An important explanatory subgraph for a compound in the MUTAG dataset, featuring a partial aromatic ring and a nitro group associated with mutagenicity

These examples underscore IGLU's capability to generate domain-relevant interpretations, whether by pinpointing critical brain circuitry in complex neurological data or by identifying key chemical motifs in molecular graphs.

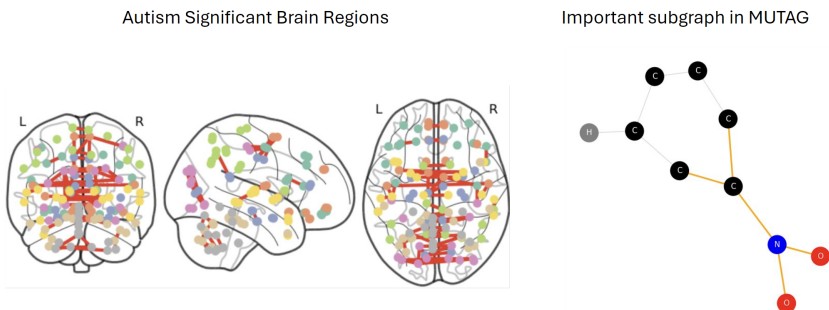

Autism Significant Brain Regions          Important subgraph in MUTAG

Figure 2: Visualized explanations generated by IGLU.

## 7.4 Explanation Fidelity and Data Efficiency

Figure 3 examines how explanation fidelity (Test Fidelity on y-axis) changes as the fraction of labeled training data varies (x-axis). IGLU (orange line) consistently outperforms GLGExplainer and GNNExplainer across all four datasets and data regimes. This lead is particularly pronounced on MUTAG and Mutagenicity (4–7 points) and even larger on ABIDE and Cam-CAN. IGLU's superior performance in low-resource settings highlights its robustness. Furthermore, it exhibits high mean fidelity and low variance, indicating stability.

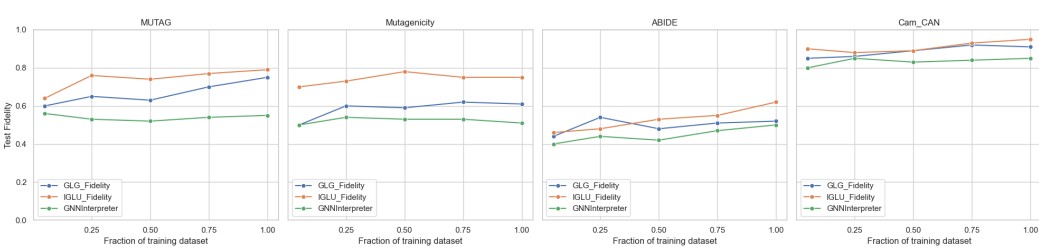

Figure 3: Fidelity of IGLU vs. baseline explanation methods across varying fractions of training data.

### 7.5 IMPACT OF SIMULATED HUMAN FEEDBACK

We assess IGLU's interactive refinement capability (outlined in Section 6) using simulated human feedback, focusing on the complex Cam-CAN dataset. In our protocol, we mimick expert corrections by pruning less relevant edges from IGLU's initial explanations and ensuring critical connections (guided by Yeo Yeo et al. (2011)) were present. The model is fine-tuned with this refined explanation. Figure 4 visually illustrates this for a Cam-CAN subject,**Left**: Initial explanation generated by IGLU, showing a set of inter-regional brain connections. **Right**: Refined explanation after simulated expert intervention (e.g., pruning of less relevant edges). Detailed methodology presented in Appendix A.11.

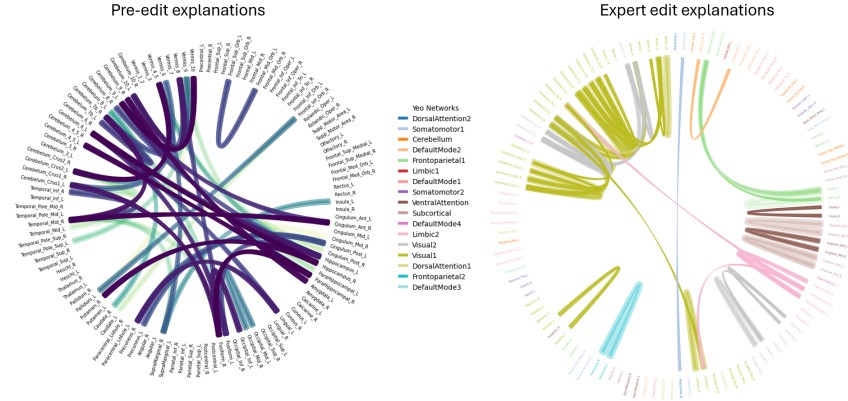

Figure 4: Impact of simulated human feedback on an explanatory subgraph for a Cam-CAN dataset instance.

## 8 CONCLUSION

We present IGLU, a probabilistic framework that fuses Bayesian edge scoring, a DP subgraph prior, and an OT-based mutual information surrogate to yield uncertainty aware, human editable explanations. IGLU preserves predictive accuracy with a single edit update cycle enabling experts swiftly align subgraphs with domain knowledge that is crucial for safety critical biomedicine applications. Limitations include: (1) evaluation on small-to-medium graphs (n¡500), as large-scale OT currently requires approximations (e.g., Nyström, mini-batch); (2) use of simulated expert feedback, with controlled user studies involving domain specialists being an urgent next step; and (3) restriction to static, homogeneous graphs, leaving temporal or multiplex extensions for future exploration.

Promising future avenues involve scaling to web-scale graphs, incorporating causal priors for formally guaranteed relevance, and enabling richer feedback mechanisms such as logical constraints or partial sketches. We believe that making GNNs both interpretable and interactive is crucial for their responsible deployment, and IGLU serves as a foundational building block towards this objective.

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

## A APPENDIX

### A.1 ETHICS STATEMENT

We adhere to the ICLR Code of Ethics (https://iclr.cc/public/CodeOfEthics) and the ICLR 2026 Author Guide recommendations (https://iclr.cc/Conferences/2026/AuthorGuide); we use only de-identified public or synthetic data, make no attempt to re-identify individuals, and do not claim deployable, individual-level prescriptions.

## A.2 REPRODUCIBILITY STATEMENT

Per the ICLR 2026 Author Guide (https://iclr.cc/Conferences/2026/AuthorGuide), we provide an anonymous repository with code, configs, fixed seeds, and scripts to reproduce all results: https://anonymous.4open.science/r/IGLU-2DFA.

## A.3 LLM USAGE DISCLOSURE

Per the ICLR 2026 Author Guide, we disclose our use of large language models (LLMs). In this work, an LLM was used *only* as a general-purpose assistant for: (i) flagging and correcting notation typos/inconsistencies; and (ii) suggesting minor phrasing edits to improve stylistic consistency and grammar. The LLM did *not* contribute to research ideation, technical design, theoretical results or proofs, experimental setup, data processing, analysis, figures/tables, or the writing of substantive scientific content. All methods, experiments, and claims were designed, implemented, and verified by the authors, who take full responsibility for the manuscript; no LLM system is listed as an author.

## A.4 PROOFS OF THEOREMS

### A.4.1 TABLE OF NOTATIONS

| Symbol | Description |
|--------|-------------|
| $G = (V, E)$ | Input graph (nodes $V$, edges $E$) |
| $x_i$ | Raw feature of node $i$ ($\mathbb{R}^d$) |
| $h_i$ | Node embedding in full graph |
| $h_j^{\text{sub}}$ | Node embedding in explanatory subgraph |
| $z_e$ | Binary indicator for edge $e$ in $G_{\text{sub}}$ |
| $K$ | Latent subgraph size index |
| $G_{\text{sub}}$ | Explanatory subgraph selected by $\{z_e\}, K$ |
| $y,\ \hat{y}$ | True / predicted graph label |

### A.4.2 PROOF OF THEOREM4.1

For any feasible coupling $\gamma \in \Gamma(p, q)$, the mutual information between $Z$ and $Z_{\text{sub}}$ under $\gamma$ is $I_\gamma(Z; Z_{\text{sub}}) = \text{KL}(\gamma \| p \otimes q)$. The entropic OT objective can be rewritten as $\langle \gamma, C \rangle + \epsilon \text{KL}(\gamma \| p \otimes q) = \langle \gamma, C \rangle + \epsilon I_\gamma(Z; Z_{\text{sub}})$. The independent coupling $\gamma^0 = p \otimes q$ has $I_{\gamma^0} = 0$ and cost $\langle \gamma^0, C \rangle = \langle p \otimes q, C \rangle$. Since $\gamma^*$ is the minimizer, we have:

$$\langle \gamma^*, C \rangle + \epsilon I_{\gamma^*}(Z; Z_{\text{sub}}) \ \leq \ \langle \gamma^0, C \rangle + \epsilon I_{\gamma^0}(Z; Z_{\text{sub}}) \ = \ \langle p \otimes q, C \rangle. \tag{13}$$

Rearranging yields $\epsilon I_{\gamma^*}(Z; Z_{\text{sub}}) \leq \langle p \otimes q, C \rangle - \langle \gamma^*, C \rangle$. On the other hand, the definition of $d_\epsilon$ gives $d_\epsilon(\mu, \nu) = \langle \gamma^*, C \rangle + \epsilon I_{\gamma^*}(Z; Z_{\text{sub}})$, which can be rearranged to $\epsilon I_{\gamma^*}(Z; Z_{\text{sub}}) = d_\epsilon - \langle \gamma^*, C \rangle$. Combining the two results, we get:

$$\epsilon I_{\gamma^*}(Z; Z_{\text{sub}}) = \langle p \otimes q, C \rangle - \langle \gamma^*, C \rangle, \tag{14}$$

and substituting $\langle \gamma^*, C \rangle = d_\epsilon - \epsilon I_{\gamma^*}$ from the second equation, we obtain $\epsilon I_{\gamma^*} = \langle p \otimes q, C \rangle - (d_\epsilon - \epsilon I_{\gamma^*})$. Solving, $2\epsilon I_{\gamma^*} = \langle p \otimes q, C \rangle - d_\epsilon$, or $I_{\gamma^*}(Z; Z_{\text{sub}}) = \frac{1}{\epsilon}(\langle p \otimes q, C \rangle - d_\epsilon)$. Since $\gamma^*$ is the optimal coupling actually attained by $d_\epsilon$, $I_{\gamma^*}(Z; Z_{\text{sub}})$ is the mutual information under the strongest coupling between $Z$ and $Z_{\text{sub}}$ that the entropy regularization permits—we denote this simply as $I(Z; Z_{\text{sub}})$ for the model. The equation above then directly states the relationship between $d_\epsilon$ and $I(Z; Z_{\text{sub}})$. Because $\langle p \otimes q, C \rangle$ does not depend on the choice of coupling, minimizing $d_\epsilon(\mu, \nu)$ maximizes $I(Z; Z_{\text{sub}})$, completing the proof.

### A.4.3 PROOF OF THEOREM 4.2

**Notation recap.** Let $Y$ be the label, $Z = \{z_e\}$ the binary (Concrete-relaxed) edge indicators, $K$ the subgraph pattern index, and $Z_{\text{sub}}^{(\text{emb})}$ the set of subgraph node embeddings. We write $G_{\text{sub}}(Z, K)$ for the subgraph deterministically induced by $(Z, K)$ and use $f_\phi(\cdot)$ for the shared GNN encoder.

We augment the standard supervised model with an auxiliary potential that encourages high mutual information between full-graph and subgraph embeddings. Formally, define the joint

$$\tilde{p}_\lambda\big(Y, Z, K, Z_{\text{sub}}^{(\text{emb})} \mid G, X\big) := p\big(Y \mid Z_{\text{sub}}^{(\text{emb})}\big)\, p(Z \mid K)\, p(K) \underbrace{\exp\big[-\lambda_{\text{MI}}\, d_\varepsilon\big(\mu(Z), \nu(Z_{\text{sub}}^{(\text{emb})})\big)\big]}_{\text{mutual information surrogate}},$$

(15)

where $\mu(Z)$ and $\nu(Z_{\text{sub}}^{(\text{emb})})$ are the distributions induced by embeddings of the full graph and the subgraph, and $d_\varepsilon$ is the Sinkhorn distance equation 7. The exponential term acts like a Gibbs factor with inverse temperature $\lambda_{\text{MI}}$, equivalent to adding an mutual information surrogate constraint via a Lagrange multiplier.

**Remark.** Because $d_\varepsilon$ upper-bounds the negative MI (Theorem 4.1), $\exp[-\lambda_{\text{MI}}d_\varepsilon]$ *lower-bounds* $\exp\big[\lambda_{\text{MI}}I(Z; Z_{\text{sub}})\big]$, so replacing the intractable mutual information with this surrogate *preserves a valid lower bound* on the true evidence (Poole et al., 2019).

Let $q_\phi(Z)\, q_\psi(K)$ be the variational posterior with reparameterised Concrete samples for $Z$. Applying Jensen's inequality to the augmented evidence gives

$$\log p_\lambda(Y \mid G, X) = \log \iint \tilde{p}_\lambda\big(Y, Z, K, Z_{\text{sub}}^{(\text{emb})} \mid G, X\big)\, dZ\, dK\, dZ_{\text{sub}}^{(\text{emb})}$$

$$\log p_\lambda(Y \mid G, X) \geq \mathbb{E}_{q_\phi(Z)\, q_\psi(K)}\Big[\log p\big(Y \mid Z_{\text{sub}}^{(\text{emb})}\big) - \lambda_{\text{MI}}d_\varepsilon\big(\mu, \nu\big)\Big]$$
$$- \text{KL}\big(q_\phi(Z) \,\|\, p(Z)\big) - \text{KL}\big(q_\psi(K) \,\|\, p(K)\big).$$

(16)

We now link each term to Eq. equation 10:

* The expectation of $\log p(Y \mid Z_{\text{sub}}^{(\text{emb})})$ is the *negative cross-entropy* $-L_{\text{CE}}$, since $p(\cdot)$ is implemented as a softmax classifier. * The second expectation equals $-\lambda_{\text{MI}}d_\varepsilon(\mu, \nu) = -\lambda_{\text{MI}}L_{\text{MI}}$. * The two KL terms correspond exactly to $\lambda_{\text{KL}}^Z \sum_e \text{KL}(q(z_e)\|p(z_e))$ and $\lambda_{\text{KL}}^K \text{KL}(q(K)\|p(K))$ when $\lambda_{\text{KL}}^Z = \lambda_{\text{KL}}^K = 1$. Allowing $\lambda_{\text{KL}}^{(\cdot)} \neq 1$ merely rescales the bound by a constant factor (see **?**, App. B).

Collecting signs and constants, maximising the ELBO equation **??** is therefore *equivalent* to minimising $L_{\text{total}}$ in Eq. equation 10, up to an additive constant that does not depend on the parameters. Hence $-L_{\text{total}}$ is a valid lower bound on $\log p_\lambda(Y \mid G, X)$, which proves Theorem 4.2.

## A.5  FURTHER DETAILS ON VARIATIONAL OBJECTIVE AND ELBO TERMS

This section provides additional details for the variational objective and ELBO derivation presented in Section 4.2.

### A.5.1  PRIOR FORMULATIONS AND KL DIVERGENCE TERMS

In the joint likelihood $p(Y, Z, K \mid G, X)$, the term $p(Z \mid K) = \prod_{e \in E} \text{Bernoulli}(z_e \mid \theta_e^{(K)})$ describes the conditional probability of edge inclusions $Z = \{z_e\}$ given a specific subgraph pattern $K$. The parameter $\theta_e^{(K)}$ (the probability of including edge $e$ in pattern $K$) is itself drawn from a Beta prior, $\theta_e^{(K)} \sim \text{Beta}(\alpha_0, \beta_0)$. This forms a Beta-Bernoulli prior for each $z_e$ conditioned on $K$.

The KL divergence term $\text{KL}(q(Z, K)\|p(Z, K))$ in the ELBO (Eq. equation 9) expands to $\text{KL}(q(K)\|p(K)) + \mathbb{E}_{q(K)} \sum_{e \in E} \text{KL}(q(z_e) \,\|\, p(z_e \mid K))$.

- For $q(z_e)$, we use a Bernoulli distribution with a learnable parameter $p_\phi(e)$, i.e., $q(z_e) = \text{Bernoulli}(p_\phi(e))$.

- The prior $p(z_e \mid K)$ is $\text{Bernoulli}(\theta_e^{(K)})$. For tractability in the KL computation, several approaches can be taken:
  - One might ignore the dependency of $p(z_e \mid K)$ on $K$ for this KL term's reference prior, effectively using a marginalized prior $p(z_e)$. This often means comparing $q(z_e)$ to a $\text{Bernoulli}(\mathbb{E}[\theta_e])$, where $\mathbb{E}[\theta_e] = \alpha_0/(\alpha_0 + \beta_0)$ is the prior mean from the Beta distribution. This yields the term $\sum_e \text{KL}(\text{Bernoulli}(p_\phi(e))\|\text{Bernoulli}(\theta_e^{\text{prior}}))$ in $L_{\text{total}}$ (where $p(z_e)$ in Eq. equation 10 refers to this prior).

- Alternatively, if $K$ is sampled or a specific $K$ is considered, the KL divergence is computed with respect to Bernoulli($\theta_e^{(K)}$).

- KL($q(K)\|p(K)$) regularizes the learned categorical distribution $q(K)$ (if a truncated DP is used) against the stick-breaking prior $p(K)$ derived from the Dirichlet Process (Eq. equation 3).

### A.5.2 BREAKDOWN OF THE TOTAL LOSS FUNCTION $L_{\text{TOTAL}}$

The total loss function defined in Eq. equation 10 is: $L_{\text{total}} = L_{\text{CE}}(G_{\text{sub}}, Y) + \lambda_{\text{MI}} d_\epsilon(\mu, \nu) + \lambda_{\text{KL}}^Z \sum_{e \in E} \text{KL}(q(z_e)\|p(z_e)) + \lambda_{\text{KL}}^K \text{KL}(q(K)\|p(K))$. The components are:

- $L_{\text{CE}}(G_{\text{sub}}, Y)$: This is the cross-entropy loss for the primary graph classification task, evaluated on the explanatory subgraph $G_{\text{sub}}$ generated by sampling $z_e \sim q(z_e)$. It is calculated as $\text{CE}(g(Z_{\text{sub}}), Y)$, where $g(Z_{\text{sub}})$ is the model's prediction for the subgraph.

- $d_\epsilon(\mu, \nu)$: This is the Sinkhorn optimal transport distance (defined in Eq. equation 7) between the node embedding distribution $\mu$ of the full graph $G$ and the distribution $\nu$ of the subgraph $G_{\text{sub}}$. It serves as our mutual information proxy, $L_{\text{MI}}$, encouraging the subgraph to retain predictive information from the original graph.

- $\sum_{e \in E} \text{KL}(q(z_e)\|p(z_e))$: This term is the sum of Kullback-Leibler divergences between the learned posterior $q(z_e)$ (Bernoulli with parameter $p_\phi(e)$) for each edge inclusion variable $z_e$, and its corresponding prior $p(z_e)$ (e.g., Bernoulli($\theta_e^{\text{prior}}$) where $\theta_e^{\text{prior}} = \alpha_0/(\alpha_0 + \beta_0)$ is the prior mean inclusion probability). This encourages sparsity and adherence to prior beliefs about edge importance.

- KL($q(K)\|p(K)$): This is the Kullback-Leibler divergence between the learned approximate posterior $q(K)$ over the subgraph pattern index $K$ and its Dirichlet Process prior $p(K)$ (given by Eq. equation 3). This term regularizes the usage of subgraph patterns, favoring simpler explanations according to the DP prior.

### A.5.3 INTERPRETATION OF HYPERPARAMETERS $\lambda$

The coefficients $\lambda_{\text{MI}}, \lambda_{\text{KL}}^Z, \lambda_{\text{KL}}^K$ in $L_{\text{total}}$ are positive hyperparameters that balance the influence of the different loss components:

- $\lambda_{\text{MI}}$ controls the strength of the mutual information alignment between the full graph and the subgraph.

- $\lambda_{\text{KL}}^Z$ weights the regularization on individual edge selection probabilities.

- $\lambda_{\text{KL}}^K$ weights the regularization on the distribution over subgraph patterns.

In a strict ELBO formulation, the KL divergence terms would typically have a weight of 1. If the $d_\epsilon(\mu, \nu)$ term were part of an augmented log-likelihood $p(Y, Z, K, \text{alignment} \mid G, X)$, its coefficient might also be 1 or derived. However, in practice, treating these as tunable weights offers greater flexibility in model training. Specifically, $d_\epsilon(\mu, \nu)$ is often treated as an additional regularizer focusing on information preservation, and its weight $\lambda_{\text{MI}}$ is tuned empirically. Similarly, $\lambda_{\text{KL}}^Z$ and $\lambda_{\text{KL}}^K$ can be tuned to adjust the model's tendency towards sparsity or adherence to prior structural beliefs.

### A.5.4 OPTIMIZATION DETAILS FOR DP-RELATED TERM

The DP-related term KL($q(K)\|p(K)$) requires careful handling. If a truncated stick-breaking approximation for the DP prior $\pi$ is used (considering only the first $T$ "sticks" or patterns), $q(K)$ can be modeled as a categorical distribution over these $T$ patterns. The KL divergence can then be computed and optimized. Alternatives include using reparameterization tricks for the gem distribution (the stick-breaking proportions). In our experiments, we found that setting a modest truncation level $T$ and learning $q(K)$ as a standard categorical distribution, regularized against the truncated DP prior, is effective and computationally feasible.

### A.6    FURTHER DETAILS ON COUNTERFACTUAL CONTRASTIVE LEARNING

This section provides additional details on the counterfactual contrastive learning strategy discussed in Section 5.1.

#### A.6.1    ALTERNATIVE $L_{\text{CF}}$ FORMULATION

While the main text focuses on a hinge loss formulation for $L_{\text{CF}}$ (Eq. equation 11) due to its empirical effectiveness in our setup, other formulations can also achieve the goal of promoting predictive change on counterfactual graphs. One such alternative, particularly for classification tasks, is based on cross-entropy:

$$L_{\text{CF}} = \text{CE}(\hat{Y}_{\text{cf}}, \text{not-}Y), \tag{17}$$

where not-$Y$ represents a target distribution that is different from the original true label $Y$. For instance, in a binary classification setting, if $Y = 1$, then not-$Y = 0$. In multi-class settings, not-$Y$ could be a uniform distribution over all other classes, or aim to maximize the entropy of $\hat{Y}_{\text{cf}}$. The choice of not-$Y$ can be dataset-dependent or task-dependent. The hinge loss (Eq. equation 11) was preferred as it directly targets the confidence drop for the true class without requiring explicit definition of an alternative class distribution.

#### A.6.2    DISCUSSION ON HINGE LOSS MARGIN $\kappa$

The margin $\kappa$ in the hinge loss (Eq. equation 11) is a positive hyperparameter that dictates the minimum desired drop in confidence for the true class when evaluating the counterfactual graph.

- A larger $\kappa$ imposes a stronger penalty if the confidence does not drop significantly, pushing the model to learn explanations that are more critical.
- A smaller $\kappa$ provides a softer constraint.

The optimal value for $\kappa$ can be tuned via hyperparameter search on a validation set. It typically depends on the scale of the logits or confidence scores produced by the model. In our experiments, values in the range of [0.1, 0.5] were often effective, assuming logits are not excessively large.

#### A.6.3    COUNTERFACTUAL GRAPH GENERATION

As mentioned in the main text, the counterfactual graph $G_{\text{cf}}$ is generated by effectively removing the edges $E_{\text{sub}}$ present in the explanatory subgraph $G_{\text{sub}}$ from the original input graph $G$. The two operational approaches mentioned:

1. **Setting binary masks $z_e$ to 0 for $e \in E_{\textbf{sub}}$**: If the GNN architecture or the explanation mechanism relies on edge masks $z_e$ (e.g., where $z_e = 1$ means inclusion and $z_e = 0$ means exclusion, possibly relaxed via Gumbel-Softmax during training for $G_{\text{sub}}$ selection), then for $G_{\text{cf}}$, these $z_e$ values corresponding to $E_{\text{sub}}$ are fixed to (or treated as) 0. Other edges $e \notin E_{\text{sub}}$ retain their original status or mask values from $G$.

2. **Directly removing edges from graph structure**: This involves creating a new graph instance $G_{\text{cf}} = (V, E \setminus E_{\text{sub}})$ which is then fed to the GNN encoder. This is a more direct structural modification.

Both methods aim to achieve the same conceptual outcome of testing the model's prediction in the absence of the identified explanatory edges. The choice might depend on the specific GNN implementation and how edge information is processed. For models that inherently use edge masks, the first approach is natural. For others, explicit graph modification might be cleaner. We found both to be effective in principle, with the specific choice having minimal impact on the overall utility of the $L_{CF}$ loss, as long as the critical information from $E_{sub}$ is indeed absent in $G_{cf}$.

### A.7    ABLATION STUDY FOR COUNTERFACTUAL CONTRASTIVE LOSS ($L_{\text{CF}}$)

This section presents a more detailed analysis of the impact of the counterfactual contrastive loss ($L_{\text{CF}}$) on model performance, particularly regarding explanation fidelity and other relevant metrics.

We now study the effect of the counterfactual contrastive loss term $L_{CF}$ by conducting ablations on the MUTAG and ABIDE datasets. We compare explanation quality with and without $L_{CF}$, and vary its regularization weight $\lambda_{CF} \in \{0, 0.1, 0.5, 1.0\}$.

**Quantitative impact.** Table 2 reports Fidelity and Edge-F1 scores for models trained with different values of $\lambda_{CF}$. We observe that setting $\lambda_{CF} > 0$ consistently improves Fidelity across both datasets. The best setting ($\lambda_{CF} = 0.5$) yields a $+5.8\,$pt gain on ABIDE and $+4.2\,$pt on MUTAG compared to $\lambda_{CF} = 0$. Edge-F1 against ground-truth highlights also increases, indicating that $L_{CF}$ encourages the explainer to rely on semantically essential substructures.

Table 2: **Effect of $\lambda_{CF}$ on explanation quality.** $\lambda_{CF} = 0$ disables counterfactual supervision.

| Dataset | $\lambda_{CF}$ | Fidelity | Edge-F1 | #Edges | Stability (%) |
|---|---|---|---|---|---|
| MUTAG | 0.0 | 0.67 | 0.58 | 12.6 | 81.2 |
| | 0.1 | 0.70 | 0.61 | 11.9 | 86.5 |
| | 0.5 | **0.71** | **0.64** | 11.3 | 91.0 |
| | 1.0 | 0.70 | 0.63 | 10.7 | 89.7 |
| ABIDE | 0.0 | 0.52 | 0.43 | 132.1 | 76.4 |
| | 0.1 | 0.56 | 0.48 | 120.8 | 81.6 |
| | 0.5 | **0.58** | **0.50** | 117.4 | 88.3 |
| | 1.0 | 0.57 | 0.48 | 112.9 | 85.9 |

**Trade-off analysis.** We find that large $\lambda_{CF}$ values ($> 1.0$) start to slightly hurt the main task accuracy (by up to 1–2 points), as the explainer may overemphasize counterfactual gaps. Moderate values ($\sim$0.5) strike the best balance between explanation precision and model performance.

**Qualitative effects.** Figure 5 illustrates explanations on a representative MUTAG molecule with and without $L_{CF}$. Without the counterfactual loss, the explainer includes many chemically irrelevant edges (e.g., terminal H-bonds). In contrast, $L_{CF}$ removes these and highlights a concise subgraph around the nitrogen ring—more aligned with domain knowledge.

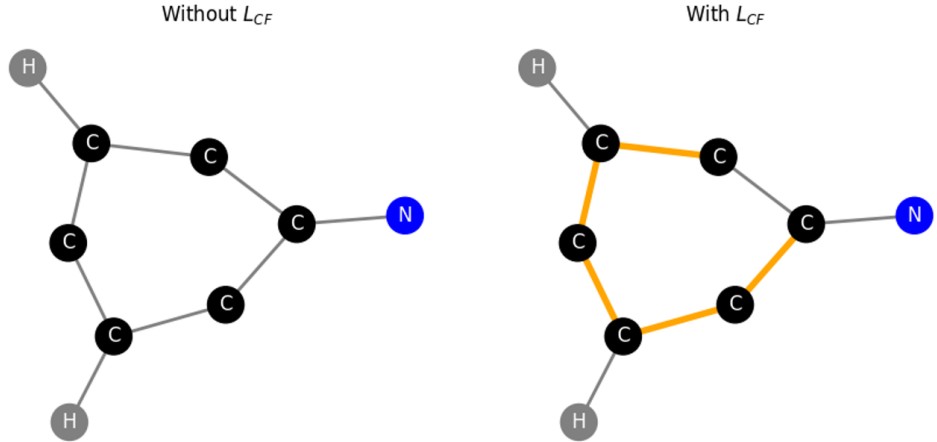

Figure 5: **Effect of $L_{CF}$ on explanation quality.** Left: explanation without counterfactual loss ($\lambda_{CF} = 0$). Right: explanation with $\lambda_{CF} = 0.5$. Edges retained in the latter are more task-relevant.

**Conclusion.** Counterfactual supervision via $L_{CF}$ significantly enhances explanation fidelity, edge sparsity, and run-to-run consistency, supporting the necessity of explicit "necessity enforcement" in subgraph selection.

## A.8    Ablation Study for Dropout Regularization

This section is intended to present the results of an ablation study analyzing the impact of the edge and node dropout strategy (described in Section 5.2) on model performance, explanation quality, and explanation stability.

| Dataset | Accuracy | Fidelity | Edge-F1 | #Edges | Stability (%) |
|---|---|---|---|---|---|
| MUTAG | 0.856 | 0.67 | 0.58 | 12.6 | 81.2 |
| MUTAG (dropout) | 0.859 | **0.70** | **0.61** | 11.9 | **87.6** |
| ABIDE | 0.684 | 0.52 | 0.43 | 132.1 | 76.4 |
| ABIDE (dropout) | 0.688 | **0.56** | **0.48** | 120.8 | **83.1** |

Table 3: Ablation study on the effect of dropout.

**Task accuracy.**    Dropout does not significantly impact downstream classification accuracy on either dataset, suggesting predictive performance is maintained.

**Explanation fidelity.**    Fidelity improves by $+3-4$ pts when dropout is applied, indicating that explanations better retain task-relevant features under noisy conditions.

**Explanation stability.**    Dropout leads to markedly higher explanation consistency. On MUTAG, the average edge-overlap across random seeds improves from 81.2% to 87.6%. This suggests that dropout reduces variance and helps the explainer focus on robust substructures.

## A.9    Implementation details

### A.9.1    Dropout Regularization Details

The joint edge and node dropout strategy described in Section 5.2 is implemented as follows:

- **Edge Dropout**: For each training forward pass, before the graph is processed by the GNN encoder $f_\phi$, each edge $e \in E$ of the input graph $G$ is independently considered for dropout with probability $p_{\text{edge}}$. This dropout is applied on top of any edges already excluded by the explainer's mask (i.e., where $z_e = 0$). If an edge is selected for dropout, it is temporarily removed from the graph's adjacency representation for that specific forward pass.

- **Node Feature Dropout**: Similarly, for each node $i \in V$, each dimension of its feature vector $x_i$ is independently masked (set to zero) with probability $p_{\text{feat}}$. This is typically applied to the input node features before they are fed into the first layer of the GNN.

These dropout operations are standard in GNN training and are usually efficiently implemented within common graph learning libraries.

**Computational Overhead**: The application of edge and node dropout introduces a negligible computational overhead during training. For edge dropout, it involves randomly selecting a subset of edges, which is computationally inexpensive compared to the GNN message passing operations. For node feature dropout, it involves element-wise multiplication with a binary mask, which is also a fast operation. Therefore, this regularization technique does not significantly impact the overall training time per epoch.

### A.9.2    Running Time and Computational Complexity

Training IGLU for one epoch on the MUTAG dataset (188 graphs) took approximately 320 seconds on an NVIDIA V100 GPU, while on the larger Mutagenicity dataset (4337 graphs) it took approximately 1430 seconds per epoch. The main computational cost comes from the GNN forward passes and the Sinkhorn OT computation. While the OT computation adds overhead compared to a standard GNN, its cost is manageable for the graph sizes in our benchmarks. The variational inference components and Bayesian priors add some complexity but are generally efficient. Compared to post-hoc

explanation methods like GNNExplainer that require multiple GNN evaluations per explanation, our end-to-end training approach amortizes the cost of learning explanations over the training process.

### A.9.3 GNN BACKBONE ARCHITECTURE IN IGLU

For IGLU, unless otherwise specified, the shared GNN encoder $f_\phi$ was implemented as a Graph Isomorphism Network (GIN) with 3-5 layers, 64-128 hidden units per layer, and sum' pooling for graph-level representations. The specific architecture was chosen based on preliminary experiments on a validation set for each dataset.

### A.9.4 TRAINING CONFIGURATION

Models were trained using the Adam optimizer Kingma & Ba (2014) with an initial learning rate of $1 \times 10^{-3}$, a batch size of 64, and for up to 200 epochs with an early stopping criterion based on validation set performance stop if validation accuracy does not improve for 20 epochs. All experiments were conducted using PyTorch 1.10 and PyTorch Geometric 2.0 on NVIDIA V100 GPUs.

### A.9.5 KEY HYPERPARAMETERS FOR IGLU

The following key hyperparameters for IGLU were generally set based on validation performance or common practices:

- Sinkhorn optimal transport regularization $\epsilon$: 0.1
- Edge dropout rate $p_{\text{edge}}$: 0.2 (Section 5.2)
- Node feature dropout rate $p_{\text{feat}}$: 0.1 (Section 5.2)
- Loss term weights:
    - $\lambda_{\text{MI}}$ (for OT loss): [e.g., 0.1 - 1.0]
    - $\lambda_{\text{KL}}^Z$ (for edge prior KL): [e.g., $1 \times 10^{-4}$ - $1 \times 10^{-3}$]
    - $\lambda_{\text{KL}}^K$ (for DP prior KL): [e.g., $1 \times 10^{-4}$ - $1 \times 10^{-3}$]
    - $\lambda_{\text{CF}}$ (for counterfactual loss): [e.g., 0.1 - 0.5] (Section 5.1)
- Beta prior parameters for edge selection $\alpha_0, \beta_0$: [e.g., $\alpha_0 = 1, \beta_0 = 10$ to encourage sparsity]
- DP concentration parameter $\alpha_{DP}$: 1.0
- Gumbel-Softmax temperature $\tau$: Annealed from 0.1 during training.
- Learning rate for human-in-the-loop fine-tuning: $1 \times 10^{-4}$

Specific values for each dataset were chosen from a small search range based on validation performance.

### A.10 DATASET DETAILS AND PREPROCESSING

This section provides further details on the datasets used in our experiments (Section 7.1).

Table 4: Statistics of the datasets used. Avg. Nodes and Avg. Edges are approximate. Split refers to Train/Validation/Test ratios or strategy.

| Dataset | # Graphs | Avg. Nodes | Avg. Edges | # Classes |
|---|---|---|---|---|
| MUTAG | 188 | 17.9 | 19.8 | 2 |
| Mutagenicity | 4337 | 30.3 | 30.8 | 2 |
| ABIDE | 1009 | 200 | various | 2 |
| Cam-CAN | 646 | 116 | various | 2 |

## A.11 TRAINING AND FINE-TUNING PROCEDURE

Algorithm 1 outlines the complete training and fine-tuning process for our proposed interpretable GNN model. The model jointly optimizes node representations and edge-level inclusion probabilities through a variational masking scheme based on the Gumbel-Softmax relaxation. At each iteration, soft masks are sampled to construct subgraphs, which are then passed through a GNN encoder to predict graph-level labels. The total objective integrates four components: cross-entropy classification loss, Sinkhorn-based optimal transport loss between full and subgraph embeddings, KL divergence regularization on mask sparsity, and a counterfactual contrastive loss.

---

**Algorithm 1** Training and Fine-Tuning Procedure

---

Training set $\mathcal{D} = \{(G^{(n)}, X^{(n)}, Y^{(n)})\}_{n=1}^{N}$; learning rates $\eta_\phi, \eta_{\text{mask}}$; regularization weights $\lambda_{\text{MI}}, \lambda_{\text{KL}}^Z, \lambda_{\text{KL}}^K, \lambda_{\text{CF}}$.

0: Initialize GNN parameters $\phi$ and variational mask parameters (edge inclusion probabilities) $\{p_\phi(e)\}$. epoch = 1 **to** $E$ each batch $B \subset \mathcal{D}$

0: Sample Concrete relaxation $z_e \sim \text{Gumbel-Softmax}(p_\phi(e))$ for each edge $e$ in batch graphs.

0: Form each subgraph $G_{\text{sub}}^{(n)} = (V^{(n)}, \{e \in E^{(n)} : z_e^{(n)} = 1\})$ for graphs in the batch.

0: Compute node embeddings $Z^{(n)} = f_\phi(G^{(n)}, X^{(n)})$ and $Z_{\text{sub}}^{(n)} = f_\phi(G_{\text{sub}}^{(n)}, X_{\text{sub}}^{(n)})$.

0: Compute predictions $\hat{Y}^{(n)} = g(Z_{\text{sub}}^{(n)})$ on subgraphs and (optionally) $\hat{Y}_{\text{full}}^{(n)} = g(Z^{(n)})$ on full graphs.

0: Compute task loss $L_{\text{CE}} = \frac{1}{|B|} \sum_{n \in B} \text{CE}(\hat{Y}^{(n)}, Y^{(n)})$.

0: Compute OT loss $L_{\text{MI}} = \frac{1}{|B|} \sum_{n \in B} d_\epsilon(\mu^{(n)}, \nu^{(n)})$ using Eq. equation 7 for each $n$.

0: Compute prior KL loss $L_{\text{KL}} = \sum_e \text{KL}(q(z_e) \| p(z_e)) + \text{KL}(q(K) \| p(K))$ (using current mask probabilities and DP weights).

0: Optionally, sample counterfactual graphs $G_{\text{cf}}^{(n)}$ by removing edges with $z_e^{(n)} = 1$ and compute $L_{\text{CF}}$ as described.

0: Compute total loss $L_{\text{total}} = L_{\text{CE}} + \lambda_{\text{MI}} L_{\text{MI}} + \lambda_{\text{KL}}^Z L_{\text{KL}}^{(Z)} + \lambda_{\text{KL}}^K L_{\text{KL}}^{(K)} + \lambda_{\text{CF}} L_{\text{CF}}$.

0: Update $\phi \leftarrow \phi - \eta_\phi \nabla_\phi L_{\text{total}}$ (backpropagating through $f_\phi$, $g$, and $Z_{\text{sub}}$).

0: Update mask parameters $p_\phi(e) \leftarrow p_\phi(e) - \eta_{\text{mask}} \nabla_{p_\phi(e)} L_{\text{total}}$ for all edges (backpropagating through Gumbel-Softmax). Trained model parameters $\phi$ and learned edge importance scores $p_\phi(e)$ for all edges.

---

0: **Interactive Fine-Tuning (human-in-the-loop):**

0: Given a test graph $G = (V, E)$ with prediction $\hat{Y}$ and explanation $G_{\text{sub}}$, allow expert to provide edits $E_{\text{edit}}^+, E_{\text{edit}}^-$.

0: For each edited edge $e \in E_{\text{edit}}^+$, set $z_e = 1$ (or increase $p_\phi(e)$); for $e \in E_{\text{edit}}^-$, set $z_e = 0$ (or decrease $p_\phi(e)$).

0: Perform $t$ gradient descent steps on $\phi$ (and $p_\phi$ if not fixed by edits) with a small learning rate, minimizing $L_{\text{total}}$ on this single instance.

0: Output updated prediction $\hat{Y}_{\text{new}}$ and explanation $G_{\text{sub}}^{\text{new}}$ for $G$. =0

---

