# OpenReview forum: "Interactive and Explainable  Graph Neural Networks with Uncertainty Awareness and Adaptive Human Feedback"
_ICLR.cc/2026/Conference — ICLR 2026 Conference Withdrawn Submission_

### Official Review · Reviewer_puev · 2025-10-16

**Soundness:** 2
**Presentation:** 1
**Contribution:** 2
**Rating:** 2
**Confidence:** 3

**Summary:**

The paper proposes IGLU, a unified probabilistic framework for interactive and explainable graph neural networks that integrates uncertainty-aware edge selection, adaptive subgraph sizing, and human-in-the-loop refinement. Core components include: Beta–Bernoulli edge inclusion with sparsity-friendly priors, a Dirichlet Process stick-breaking prior to adapt subgraph size per instance, and a Sinkhorn optimal-transport alignment used as a tractable surrogate for mutual information between full-graph and subgraph embeddings. The training objective combines cross-entropy, OT-based alignment, and KL regularization terms, with an auxiliary counterfactual contrastive loss to encourage necessity of selected edges. Experiments on MUTAG, Mutagenicity, ABIDE, and Cam-CAN report competitive accuracy and sensitivity versus standard GNN backbones and specialized neuroimaging GNNs, plus higher explanation fidelity across data regimes.

**Strengths:**

1. Integrates Bayesian edge scoring, DP-based adaptive subgraph sizing, and OT alignment into a differentiable objective
2. Human-in-the-loop editing is an important direction for explainable GNNs
3. Evaluations span chemistry and neuroimaging, demonstrating applicability across diverse graph types

**Weaknesses:**

1. Theorem 4.1 posits a link between Sinkhorn distance and mutual information, but no sensitivity analysis or ablation assesses whether maximizing this surrogate improves downstream prediction fidelity or explanation quality in practice. Coefficients such as λ_MI are not studied in the main text.
2. The framework emphasizes uncertainty-aware edge inclusion, yet there are no calibration metrics (ECE, Brier, coverage) or reliability diagrams for edge probabilities or explanation confidence
3. Refinement is demonstrated on Cam-CAN using simulated expert edits informed by literature (Yeo atlas), without user studies or quantitative measures (task time, consistency, trust) to support claims of interactive utility
4. Accuracy and sensitivity improvements over strong backbones (GIN, GAT, CIN++, IBGNN/BrainGNN) are small and dataset-dependent, which weakens claims of broad superiority
5. Explanation fidelity needs clearer definition and computation details in the main paper for reproducibility and fair comparison

**Questions:**

1. How calibrated are edge inclusion probabilities and explanation confidence scores?
2. Clearly define the fidelity metric in the main text

---

### Official Review · Reviewer_FyPE · 2025-10-29

**Soundness:** 2
**Presentation:** 2
**Contribution:** 2
**Rating:** 2
**Confidence:** 4

**Summary:**

This paper proposed IGLU, a probabilistic GNN framework that unifies uncertainty estimation and human feedback. It used a latent-variable model to assign sparse edge importance and adapt subgraph size, ensuring fidelity through a differentiable alignment objective. Experts can interactively refine explanations, with feedback dynamically integrated to improve future inferences and maintain interpretability.

**Strengths:**

(S1) IGLU unified Bayesian edge scoring, Dirichlet process priors, and optimal transport alignment into an end-to-end probabilistic GNN explainability framework.

(S2) It enabled real-time expert interaction through editable subgraph explanations with adaptive uncertainty updates.

(S3) Experiments showed consistent accuracy and high explanation fidelity across multiple graph benchmarks.

**Weaknesses:**

(W1) The paper lacks comparisons with recent methods, as the latest explanation baseline of explanation is from 2023 and the GNN competitors are outdated, making it difficult to validate the claimed improvements. Moreover, since the approach requires expert intervention to refine explanations, its practicality for application is limited.

(W2) The motivation is insufficiently articulated, particularly regarding the rationale for incorporating human feedback.

(W3) The results require deeper interpretation and an ablation study (excluding L_CF) to clarify the contributions of each component.

(W4) The writing should be significantly improved. For example, the notations in the overview figure are inconsistent with those in the manuscript (e.g., G1 in Figure 1 vs. G_sub^((k)) in line 144). There are also typos (e.g., lines 71, 479) that should be carefully corrected, and spaces should be added before citations to enhance readability.

(W5) Although a code link is provided, the referenced code files are missing or inaccessible.

**Questions:**

1. What is the specific design or novelty of the expert feedback mechanism? If other methods also allow experts to modify graphs by adding or removing edges, how does IGLU’s interactive fine-tuning differ?

2. In Figure 2 (left), what do the visualization results represent? The figure shows many colored nodes--do the colors have specific meanings?

3. Is contrastive learning essential to the framework, given that L_CF is missing from the overview figure?

4. What do the results in Figure 4 represent, and how should the figure be interpreted? A clearer explanation is needed.

5. Why no results for BrainGNN and IBGNN on MUTAG and Mutagenicity datasets?

6. How are the predictions from the subgraph and full graph combined or handled during training, since the overview figure only shows prediction from the subgraph?

---

### Official Review · Reviewer_UMxp · 2025-11-01

**Soundness:** 2
**Presentation:** 2
**Contribution:** 3
**Rating:** 4
**Confidence:** 4

**Summary:**

This paper proposes IGLU, a unified framework for generating interactive and uncertainty-aware explanations for Graph Neural Networks (GNNs). IGLU models edge importance via a Beta-Bernoulli prior, enabling calibrated confidence estimates, and employs a Dirichlet Process (DP) to dynamically control the size of the explanatory subgraph per instance. To ensure the selected subgraph preserves predictive fidelity, it introduces a Sinkhorn Optimal Transport (OT) loss aligning the embeddings of the subgraph and the full graph. Furthermore, IGLU supports human-in-the-loop editing, allowing domain experts to add/remove edges in explanations, which are then integrated into the model via lightweight fine-tuning. Empirical results on chemical and neuroimaging graph datasets demonstrate IGLU’s superiority in producing faithful, concise explanations while maintaining competitive predictive accuracy. Simulated expert feedback confirms the effectiveness of its interactive refinement mechanism.

**Strengths:**

1. Combines Bayesian uncertainty modeling with Dirichlet Process priors and Sinkhorn transport loss in a novel, coherent framework for GNN explainability.

2. Allows users to interactively edit the explanation subgraph, and incorporates this feedback efficiently during training with few gradient updates.

3. Demonstrates applicability on real-world and synthetic datasets, including neuroimaging, where interpretability is crucial.

4. Outperforms baselines in low-data regimes and under interactive correction scenarios.

**Weaknesses:**

1. Text in some figures (e.g., Figures 1, 2, and experimental plots) is too small, making them hard to interpret. Also, variable notation (e.g., whether Z refers to edge mask or embedding) is occasionally ambiguous.

2. Important recent baselines like GSAT (Stochastic Attention) and ConfExplainer (Confidence-Aware Explainer) are missing. Also, commonly used explanation metrics (e.g., AUC for edge recovery) are not reported.

3. The total loss has several components. However, no sensitivity analysis is provided for key hyperparameters (e.g., $\alpha_{DP}, \alpha_0, \beta_0$). Ablation studies isolating the contribution of each loss term would be informative.

4. While an anonymous code link is provided, the actual code repository appears empty or inaccessible, limiting reproducibility.

**Questions:**

1. The paper uses edge inclusion probability $p(e)$ as an estimate of confidence. Should the variance or posterior entropy also be considered for a more complete uncertainty quantification?

2. Section 4.1 introduces an OT-based surrogate for MI via embedding alignment. However:

2.1 Why is CE loss alone insufficient to ensure predictive information retention from the original graph?

2.2 How does aligning embeddings of $G$ and $G_{\text{sub}}$ specifically mitigate this?

2.3 Could this alignment strategy (especially without hard binarization) encourage copying the full graph? How is sparsity explicitly enforced?

3.  The interactive refinement component is promising but underexplained:

3.1 Which model parameters are updated during editing?

3.2 Why choose fine-tuning over, say, reinforcement learning or constraint-based approaches?

3.3 How does this compare to existing interactive GNNs (e.g., Interactive GNNs or IGNN frameworks)?

---

### Official Review · Reviewer_kivc · 2025-11-04

**Soundness:** 2
**Presentation:** 2
**Contribution:** 3
**Rating:** 2
**Confidence:** 3

**Summary:**

In this paper, the authors introduce IGLU, an interpretable GNN framework. Alongside the task, the model also learns how to sample the edges required to correctly perform the prediction, allowing it to assign importance to the edges depending on their sampling probability. IGLU is tested both in terms of performance and the fidelity of the explanations with respect to the dataset ground truth. Moreover, the model allows for human feedback, and the experiments showcase this feature as well.

**Strengths:**

- The authors introduce an interesting and grounded framework for developing an interpretable GNN

- Fundamentally, the model learns a probability distribution over the edges to sample through a Bayesian inference method, with additional, justifiable regularisations, providing a clear interpretation of the importance assigned by the model to the edges

- The framework allows for human intervention and feedback as well, allowing for correcting the model predictions along the way

- It does not show any significant accuracy-interpretability trade-off

**Weaknesses:**

The most critical weakness is that Section 7.5 only presents "simulated human feedback" rather than actual user studies with domain experts. The simulation protocol assumes access to ground truth knowledge, which defeats the purpose of interactive refinement in real scenarios. The authors acknowledge this limitation in Section 8, but this severely weakens the paper's central claim about "adaptive human feedback". For a paper emphasizing human-in-the-loop capabilities in the title and abstract, the absence of genuine human evaluation is a major deficiency that questions the practical utility of the interactive component.

Other weaknesses:

1. There is no code allowing for the reproducibility of the experiments, as following the repository leads to a repository where none of the files are available.

2. The title talks about “uncertainty awareness”, and the abstract of “calibrated uncertainty estimate”. However, these terms generally refer in the literature to a model capable of providing a calibrated probability (e.g., Minderer, Matthias, et al. "Revisiting the calibration of modern neural networks." NeurIPS 2021), which is something that IGLU does not do.

3. There are many formatting and clarity issues, some of which I mention below

4. The text at line 237 mentions Theorem 4.1, which is missing its header. Moreover, its proof in the appendix of the theorem is wrong, as in lines 690-691 $\epsilon I_\gamma = \langle p \otimes q, C \rangle - (d - \epsilon I_\gamma)$ clearly implies $0 = \langle p \otimes q, C \rangle - d$ and not $2 \epsilon I_\gamma = \langle p \otimes q, C \rangle - d $, as the writers suggest. It is not clear how the theorem result justifies the framework.

5. The edge sampling parameter $z_e$ ultimately depends on a function $p_\Phi (e)$, which is not clear whether it is computed from the model node embedding or with an auxiliary network in the experiments.

6. The authors do not provide any related work related to interpretable GNN models or pointers to the literature. One of the interpretable models they use, GLGExplainer, is cited wrong as “Jindi Wang, Wenbing Lyu, Xinyi Chen, Peng Cui, and Wenwu Zhu. Glgexplainer: A geometric and logical graph explainer., ICML 2023”. However, this paper does not exist, and the only GLGExplainer method I found in the literature is from Azzolin, S., Longa, A., Barbiero, P., Lio, P., And Passerini, A. Global Explainability of GNNs via Logic Combination of Learned Concepts. ICLR 2023. This raises doubts regarding the bibliography exactness.

7. It is not clear how the difference in fidelity between IGLU and GNNExplainer/GLGExplainer is larger in the ABIDE and Cam-can datasets than in MUTAG and Mutagenicity as the authors suggest in line 430, since the plots suggest it is less pronounced.

8. The citations should almost always be in parentheses, as per the ICLR template.

**Questions:**

How do you think your model is related to the calibration and uncertainty estimation literature?

How would you justify the lack of accuracy-interpretability trade-off of your framework? Are there any insights on how using only information from a selected subgraph does not impact the accuracy of your models?

I see that you perform a dropout ablation study in the appendix; however, the accuracies with dropout are not consistent with the values reported in the main paper. How is that so? Did you train all the baselines using the dropout, or was it something that you used only for IGLU?

Could you provide additional ablation studies on the impact of the various terms in your loss?

How do you compute p_\Phi (e) (see the weakness n.3) in your experiments?

---

### Note · Authors · 2025-11-28

I have read and agree with the venue's withdrawal policy on behalf of myself and my co-authors.